# Laser-Induced Porcine Model of Experimental Retinal Vein Occlusion: An Optimized Reproducible Approach

**DOI:** 10.3390/medicina59020243

**Published:** 2023-01-27

**Authors:** Mads Odgaard Mæng, Nirrooja Roshanth, Anders Kruse, Jonas Ellegaard Nielsen, Benedict Kjærgaard, Bent Honoré, Henrik Vorum, Lasse Jørgensen Cehofski

**Affiliations:** 1Department of Ophthalmology, Aalborg University Hospital, 9000 Aalborg, Denmark; 2Department of Clinical Biochemistry, Aalborg University Hospital, 9000 Aalborg, Denmark; 3Biomedical Research Laboratory, Aalborg University Hospital, 9000 Aalborg, Denmark; 4Department of Clinical Medicine, Aalborg University, 9000 Aalborg, Denmark; 5Department of Biomedicine, Aarhus University, 8000 Aarhus C, Denmark; 6Department of Ophthalmology, Odense University Hospital, 5000 Odense C, Denmark; 7Department of Clinical Research, University of Southern Denmark, 5000 Odense C, Denmark

**Keywords:** retina, retinal vein occlusion, proteomics, proteome, mass spectrometry, animal model, Western blot, immunohistochemistry, experimental

## Abstract

Retinal vein occlusion (RVO) is a frequent visually disabling condition. The management of RVO continues to challenge clinicians. Macular edema secondary to RVO is often recurrent, and patients typically require intravitreal injections for several years. Understanding molecular mechanisms in RVO is a key element in improving the treatment of the condition. Studying the molecular mechanisms in RVO at the retinal level is possible using animal models of experimental RVO. Most studies of experimental RVO have been sporadic, using only a few animals per experiment. Here, we report on 10 years of experience of the use of argon laser-induced experimental RVO in 108 porcine eyes from 65 animals, including 65 eyes with experimental branch retinal vein occlusion (BRVO) and 43 eyes with experimental central retinal vein occlusion (CRVO). Reproducibility and methods for evaluating and controlling ischemia in experimental RVO are reviewed. Methods for studying protein changes in RVO are discussed in detail, including proteomic analysis, Western blotting, and immunohistochemistry. Experimental RVO has brought significant insights into molecular changes in RVO. Testing intravitreal interventions in experimental RVO may be a significant step in developing personalized therapeutic approaches for patients with RVO.

## 1. Introduction

Retinal vein occlusion (RVO) is the second most frequent retinal vascular disease after diabetic retinopathy. RVO is a prevalent cause of macular edema and ischemic maculopathy, which are often associated with significant visual loss [1,2]. Major risk factors of RVO include hypertension, increasing age, diabetes mellitus, peripheral artery disease, and glaucoma. In more than half of cases, the patient is older than 65 years at onset [3,4,5].

Based on the site of occlusion, RVO is classified into central retinal vein occlusion (CRVO) and branch retinal vein occlusion (BRVO). Hemiretinal vein occlusion (HRVO) is considered a variant of CRVO [6,7]. The management of RVO continues to challenge clinicians and the health care system. Macular edema is often recurrent, and clinical trials have revealed that approximately 40 to 60% of patients need anti-vascular endothelial growth factor (VEGF) injections four years after being diagnosed with RVO [8]. The high frequency of treatments and follow-up visits is a well-established cause of patient dropout, leading to undertreatment of macular edema [9] with suboptimal gain in visual acuity and increased risk of developing neovascularizations and neovascular glaucoma [10,11]. The number of intravitreal anti-VEGF injections for the treatment of macular edema secondary to RVO continues to grow, as documented in a recent retrospective study which found that yearly injections in Denmark increased from 3700 to 7500 treatments between 2014 and 2022 [12]. Intravitreal injections constitute a significant burden for patients in terms of frequent visits to retina clinics and are associated with high levels of anxiety and discomfort regardless of the number of previously received injections [13,14,15].

Understanding molecular mechanisms in RVO is a key element in improving the management of the condition. However, retinal tissue with RVO is generally only available from animal models as retinal tissue from human eyes with RVO is only obtainable postmortem, and animal models are necessary to study molecular changes in RVO at retinal level [16].

Experimental RVO is effectively studied using a number of techniques which are well suited for providing insights into the complex molecular mechanisms that are activated in RVO [17,18,19]. Consistent with observations in studies of ocular fluids from patients with RVO, proteomic analyses have demonstrated that RVO is associated with a multi-faceted pathological response [20,21]. The overall objective of proteome studies is to identify and quantify the entire set of proteins in a given cell, tissue, or biofluid to provide insights into the biological processes in the disease or intervention under study [16,22,23,24]. Studying interventions in experimental RVO may elucidate mechanisms of action which can potentially be used to improve existing therapies [25,26].

Previous studies of experimental RVO have generally been based on limited numbers of animals [27]. The reproducibility of RVO models is rarely addressed, as most studies of experimental RVO are short in duration, not involving follow-up studies or the testing of interventions. Here, we report on experience based on animal studies conducted over a period of ten years, including 65 animals. We report on how experimental RVO is best induced in porcine models of experimental BRVO and CRVO, generating retinal changes similar to RVO observed in the human eye. We describe in detail how proteome analysis, immunohistochemistry, and Western blots are successfully performed to analyze molecular changes in retinal tissue with experimental RVO.

## 2. Materials and Methods

We reviewed animal studies of experimental RVO performed in our laboratories in a 10-year period [25,26,28,29,30,31,32,33]. A total of 130 eyes from 65 animals were used (Table 1), which included 65 eyes with experimental BRVO from 38 animals and 43 eyes with experimental CRVO from 27 animals.

The animal experiments were approved by the Danish Animal Inspectorate and performed according to the obtained permissions (permission numbers 2013-15-2934-00775, 2016-15-0201-00971, and 2019-15-0201-01651).

### 2.1. Animal Housing and Anesthesia

Both Danish Landrace pigs and Göttingen minipigs can be used as animal models of experimental RVO. Our group recommends Danish Landrace pigs for financial as well as anatomical reasons (Table 2). Göttingen minipigs are substantially more expensive than Danish Landrace pigs. Furthermore, we find enucleation more difficult in Göttingen minipigs due to their fairly deep orbital cavities. For studies of experimental RVO, we recommend female Danish Landrace pigs of approximately 20 kg [28,34]. Animals were housed under a 12 h light/dark cycle. Anesthesia was effectively performed using an intramuscular injection of 5 mL of Zoletil (ketamine 6.25 mg/mL, tiletamine 6.25 mg/mL, zolazepam 6.25 mg/mL, butorphanol 1.25 mg/mL, and xylain 6.25 mg/mL).

Topical anesthesia and dilation of the pupils are successfully performed using eye drops in common clinical use in concentrations for adults. Topical anesthesia is effectively performed using oxybuprocaine hydrochloride 0.4% (Bausch and Lomb, Rochester, NY, USA) and tetracaine 1% (Bausch and Lomb), then dilation is performed using tropicamide 0.5% (Mydriacyl; Bausch and Lomb) and phenylephrine 10% (Metaoxidrin; Bausch and Lomb). To prevent corneal surface desiccation and compromised view of the retina, sodium chloride 9 mg/mL (Fresenius Kabi, Bad Homburg, Germany) is applied regularly during the experimental procedures. 

### 2.2. Experimental Retinal Vein Occlusion

#### 2.2.1. Experimental Branch Retinal Vein Occlusion

Experimental BRVO is induced close to the optic nerve head using a standard argon laser (532 nm), delivered by indirect ophthalmoscopy using a 20D lens. The laser energy is set to 400 mW with an exposure time of 550 ms. A total of 30–40 laser applications are used per occlusion. The laser is first applied along the retinal branch vein to narrow the vessel and then directly onto the vein. We recommend the creation of patches of continuous laser rather than laser with spacing between the applications. The laser applications should be whitish, but snow-whitish applications should be avoided, as excessively high energy can lead to the development of a choroidal neovascularization. Experimental BRVO is considered successful when stagnation of venous blood and development of flame-shaped hemorrhages are observed. The visual streak of the pig is affected if BRVO is induced in a superior branch retinal vein close to the optic nerve head. The opposite eye may be used as a control by creating an identical area of laser burns in a similar area, but without inducing occlusion.

#### 2.2.2. Experimental Central Retinal Vein Occlusion

CRVO is induced by applying laser at the border of the optic nerve head. Using the laser settings described above for experimental BRVO, the argon laser is applied to 3–4 retinal veins at the border of the optic nerve head. To create a successful experimental CRVO, the occlusions must be balanced and partial. Complete occlusion of four branch retinal veins results in severe retinal ischemia. By applying the laser directly to retinal veins close to the optic nerve head, thrombotic material is directed toward the optic nerve head and the lamina cribrosa, thereby preventing the generation of complete occlusions. Experimental CRVO is considered successfully induced when the stagnation of venous blood and development of flame-shaped hemorrhages appear in all retinal quadrants.

The opposite eye may be used as a control eye by creating identical areas of laser burns using similar amounts of energy, but without inducing occlusion. This can be achieved by applying the laser to areas close to the optic nerve head that are devoid of any major vessels.

### 2.3. Imaging

#### 2.3.1. Funduscopic Photography

After inducing experimental RVO, funduscopic images are obtained as documentation of successfully induced RVO. To prevent corneal surface desiccation from compromising the view of the retina, Neutral ointment (Ophtha, Teva, Søborg, Denmark) is applied before each pig is returned to housing. Funduscopic photographs are obtained within 30 min after laser-induced RVO and, thereafter, on occasions when fluorescein angiography and optical coherence tomography (OCT) are performed. Funduscopic photographs may be obtained using a portable fundus camera (Optomed Aurora, Oulo, Finland). Funduscopic photographs of experimental RVO in the porcine eye may also be obtained using common clinical equipment.

#### 2.3.2. Fluorescein Angiography

Fluorescein angiography is the gold standard for verification of successfully induced BRVO or CRVO. Fluorescein angiography is effectively performed using Heidelberg fluorescein angiography or equipment that is specifically designed for fluorescein angiography in animal models, for example RETI-map—animal (Roland Consult, Berlin, Germany) [31]. To prevent corneal surface desiccation and a compromised view of the retina, sodium chloride 9 mg/mL (Fresenius Kabi, Bad Homburg, Germany) is applied every 20 s. A peripheral venous catheter is applied to a vein in the ear of the pig, and 5 mL of fluorescein 100 mg/mL (Alcon, Geneva, Switzerland) is injected through the peripheral venous catheter, followed by 5 mL of sodium chloride 9 mg/mL (Fresenius Kabi, Bad Homburg, Germany). Fluorescein angiography is conducted for 12 min. To prevent corneal surface desiccation from compromising the view of the retina at later stages, Neutral ointment (Ophtha, Teva, Søborg, Denmark) is applied before each pig is returned to housing.

#### 2.3.3. Optical Coherence Tomography

OCT is conducted following retinal laser application. The eyes undergo anesthesia, dilation, and treatment against corneal surface desiccation, as described in Section 2.1 and Section 2.3.2. OCT scans may be obtained with equipment designed for animal research like for example RETI-map—animal (Roland Consult, Berlin, Germany). OCT scans can also be obtained using common clinical equipment, but examination tools specially designed for the use in animals are preferred.

### 2.4. Sample Collection

Enucleation can be performed under anesthesia, allowing for perfusion during enucleation. Before enucleation, the pigs are anesthetized, as previously described, using an intramuscular injection of Zoletil. The animals are euthanized immediately after enucleation. Our group recommends the collection of retinal tissue immediately after enucleation. Alternatively, the eyes can be stored at −80 °C and thawed prior to dissection. In our experience, storage at −80 °C does not affect the quality of the proteome analysis, but the eyes must be handled with care during dissection [31].

The eyes should be dissected on ice under a microscope. A circumferential incision is made around the iris, 2 mm posteriorly to the limbus, to remove the cornea and the iris. The anterior segment is removed, and the vitreous body is aspirated using a 5 mL syringe. In the eyes intended for proteomic analysis, the neurosensory retina is peeled from the retinal pigment epithelium (RPE)/choroid complex using tweezers and stored at −80 °C. In eyes intended for immunohistochemistry, complexes consisting of the neurosensory retina, the RPE/choroid complex, and the sclera are excised for analysis.

If stored at −80 °C prior to dissection, the eyes must be thawed on ice. It is important to ensure that the eyes are thawed completely, as the retina can adhere to a frozen vitreous body. Furthermore, the neuroretina must be collected gently. Otherwise, the entire retina/choroid complex may be collected, resulting in a drastic increase in tissue complexity [31].

### 2.5. Sample Preparation for Proteomic Analysis

Proteomic analysis of retinal control samples and retinal tissue exposed to RVO is effectively performed using both label-free and labeling techniques [26,28,29]. For labeling approaches, we find the 10-plex tandem mass tag (TMT) kit (Thermo Scientific, Waltham, MA, USA) useful for the study of experimental RVO. Labeling techniques are well-suited for proteome analysis of retinal tissue, and a label-free approach has also been successful in proteome analysis of ocular fluids [23].

The porcine retinal samples generally contain sufficient amounts of protein to use 100 µg of each sample for proteome analysis. A volume of an appropriate lysis buffer is added to the sample. Disulfide bonds are reduced using tris-2-carboxyethylphosphine (TCEP), and the samples are alkylated using iodoacetamide. This is followed by digestion using trypsin, as described in a previous report [33]. TMT labeling is performed as previously described [33]. Peptide samples may be purified using a Pierce C18 spin column (Thermo Scientific) and fractionated using a Pierce High pH Reversed-Phase Peptide Fractionation Kit (Thermo Scientific) as previously described [33].

Quantification using tandem-mass-tag-based mass spectrometry should be performed using appropriate settings for label-based proteomics. Our own group has performed TMT-based proteomics using the Dionex UltiMate^TM^ 3000 RSLC nanosystem coupled to an Orbitrap Fusion Tribrid mass spectrometer (Thermo Scientific, Waltham, MA, USA) equipped with an EasySpray^TM^ ion source. The peptides are separated by liquid chromatography and detected in the mass spectrometer using synchronous precursor selection. The reporter ions are detected by MS^3^ in the Orbitrap as described previously [28,29,30,33].

Raw files obtained with mass spectrometry are searched against relevant databases using MaxQuant software (Max Planck Institute of Biochemistry, Martinsried, Germany; the software can be downloaded from the following site: https://maxquant.org). In our experience, optimal proteome coverage is achieved by searching against the UniProt *Sus scrofa* database as well as the UniProt *Homo sapiens* database [28].

### 2.6. Western Blotting

Due to their large size, porcine eyes render sufficient retinal tissue for proteomic analysis as well as Western blotting. Thus, Western blotting can be utilized for a semi-quantitative validation of proteins of interest identified with proteomic analysis.

The recommended protein concentration for Western blotting is generally 1 µg/µL. Thus, 20 µg protein from each sample and 2 × Laemmli Sample Buffer (Bio-Rad Laboratories, Hercules, CA, USA) are mixed and heated for 5 min at 95 °C. Protein separation, wet blot transfer, and membrane incubation are performed as described in detail in a previous report [30]. The membranes incubate overnight at 4 °C with primary monoclonal mouse anti-β-actin antibody 1:5000 (clone AC-15, Sigma-Aldrich, St. Louis, MO, USA) as a housekeeping protein for normalization, and a primary antibody directed at the protein of interest in skim milk buffer. After incubation, relevant secondary antibodies are added as described in detail in a previous report [30], and detection of protein bands is performed using chemiluminescence, Lumi-Light, with Western Blotting Substrate (Roche Diagnostics, Indianapolis, IN, USA). Imaging and densitometric measurements of relative protein expression are performed as described in detail in a previous article [30]. Protein expression is normalized to β-actin expression, and densitometric data are used for statistical analysis [30].

For Western blotting of retinal tissue, in case β-actin is regulated in the material under study or cannot be used as a housekeeping protein for technical reasons, our group recommends the use of GAPDH for normalization purposes [26].

### 2.7. Immunohistochemistry

Immunohistochemistry is well-suited for elucidating the localization of key retinal proteins that are regulated in RVO. In our experience, the eyes should be reserved exclusively for immunohistochemistry, rather than using them for both immunohistochemistry and mass spectrometry.

Immunohistochemistry of retinal tissue with RVO is successfully fixated using either freshly made neutral buffered formalin or Clarke’s fixative [28,33].

#### 2.7.1. Fixation Using Formalin

For preparation of immunohistochemistry, the anterior segment and the vitreous body are removed. After removal of the anterior segment and the vitreous body, the globe is divided into four quadrants, which are placed in a relevant fixative, e.g., neutral buffered formalin or Clarke’s fixative. Each quadrant contains a complex consisting of the retina, the choroid, and the sclera.

Complexes consisting of retina, choroid, and sclera are fixated in neutral buffered formalin fixative for 24 h. The formalin solution is removed, and the tissue is stored in a PBS solution at 4 °C until further use. For immunohistochemistry, 4 µm thick sections are cut from neutral buffered formalin and paraffin-embedded tissue blocks. The sections are installed on FLEX IHC slides (Dako; Glostrup, Denmark), dried at 60 °C, dewaxed and rehydrated using a graded ethanol, and washed using a 0.05 M Tris-buffered saline (Fagron Nordic A/S, Copenhagen, Denmark).

Endogenous biotin reactivity and optimal epitope retrieval is performed as described in detail in a recent report [26]. Sections are incubated for 60 min using antibodies diluted in TNT Antibody Diluent (Dako, Glostrup, Denmark A/S), and visualization of the antigen–antibody complex is performed as described in a previous article [26]. Using DAB as a chromogen (K3468, Dako, Glostrup, Denmark), immunostaining, counterstaining, scanning, and image acquisition are performed as described in a recent report [26].

#### 2.7.2. Fixation Using Clarke’s Fixative

Clarke’s fixative is constructed using three parts 99% ethanol and one part glacial acetic acid. The porcine eyes are dissected as described above, and four quadrants containing retina, choroid, and sclera are fixated in the fixative for 5 h at 20 °C. After fixation, the fixative is removed, and the tissue is stored in 0.1 M phosphate buffer pH 7.4 at 4 °C. The buffer is replaced daily for the first three days after fixation and then stored at 4 °C until further use.

Immunohistochemical detection including antibody incubation and processing using EnVision DAB (Dako, Glostrup, Denmark) is performed as described in detail in a previous article [30].

### 2.8. Targeted Mass Spectrometry by Selected Reaction Monitoring (SRM)

Selected reaction monitoring (SRM) is an effective tool for validation of key proteins identified using discovery proteomics. SRM may be considered in cases in which Western blotting is not sufficiently sensitive for validation of key proteins or suitable antibodies are not available for Western blotting. The reader is directed to a previous study [35] for a detailed description of SRM as a validation tool for retinal proteins.

## 3. Results

### 3.1. Funduscopic Photography

In the reviewed studies, retinal hemorrhages and dilation of the retinal veins appeared within 20 min of induction of experimental RVO (Figure 1).

### 3.2. Fluorescein Angiography

In the reviewed studies, formation of RVO was validated using fluorescein angiography (Figure 2). Fluorescein angiography showed successfully induced RVO with delayed venous filling of the occluded veins. Diversions of venous blood for drainage by adjacent veins appeared in the angiography after approximately 12 to 14 s, and peripheral leakage developed after 30 to 40 s. No recanalization of any occlusions was observed during angiography.

Fluorescein angiography showed no retinal non-perfusion in experimental BRVO. However, some degree of retinal non-perfusion in experimental CRVO was consistently observed.

### 3.3. Optical Coherence Tomography

In OCT scans of control eyes, retinal thinning was observed in areas with laser applications (Figure 3A). Increased retinal thickening was observed in porcine eyes with RVO in retinal areas drained by occluded veins (Figure 3B,C). 

### 3.4. Assessment of Retinal Stress through Characterization of Retinal Müller Cell Proteins

Among the reviewed studies, glial fibrillary acidic protein (GFAP), vimentin, and peripherin were quantified using Western blotting in six eyes with BRVO (Figure 4). GFAP, the most specific of the three markers, and peripherin were found to be significantly increased in BRVO at *p* = 0.025 and *p* = 0.0018, respectively.

Immunohistochemistry using staining for GFAP, vimentin, and peripherin (Figure 5) in the reviewed studies confirmed increased levels of GFAP, vimentin, and peripherin, consistent with retinal stress at the molecular level with swollen retinal Müller glial cells. Increased staining for GFAP was observed in the ganglion cell layer, inner plexiform layer, and outer plexiform layer. Increased staining for vimentin was observed in the inner plexiform layer and outer plexiform layer. Increased staining for peripherin was observed in the ganglion cell layer, inner plexiform layer, and outer nuclear layer.

GFAP and vimentin are intermediate filament proteins in the Müller cells in the ra-dial glia of the retina. GFAP and vimentin are upregulated in response to retinal stress [36,37,38]. Peripherin is also an intermediate filament protein, which is known to be upregulatd in response to retinal stress [39]. As seen in Figure 4 and Figure 5, a significant increase in GFAP, vimentin, and peripherin was observed in experimental BRVO suggesting significant retinal stress at the molecular level.

### 3.5. Proteomics

After filtration, a different number of proteins were successfully identified and quantified in each of the reports, as follows: 1932 proteins with 14 proteins significantly regulated with label-free quantification [25], 3559 proteins with 21 proteins significantly regulated with TMT labeling [26], 2749 proteins with 5 proteins significantly regulated with TMT labeling [28], 4013 proteins with 9 proteins significantly regulated with TMT labeling [29], 3716 proteins with 26 proteins significantly regulated with TMT labeling [30], 1974 proteins with 52 proteins significantly regulated with label-free quantification [31], 3791 proteins with 147 proteins significantly regulated with TMT labeling [32], and 2837 proteins with 237 proteins significantly regulated with label-free quantification [33].

## 4. Discussion

### 4.1. Strengths of Experimental RVO in the Porcine Eye

The size of the porcine eye is a major advantage. Due to its size, the eye renders sufficient tissue for proteomic analysis and Western blotting, and it is unnecessary to pool samples to obtain sufficient sample material.

Several animal species have been used for the studies of experimental RVO, including rodents, rabbits, cats, dogs, pigs, and non-human primates [27]. The porcine eye has a number of advantages due to similarities with the human eye, including similar size and a fully vascularized retina [40,41,42]. These features make the porcine retina well-suited for studies of experimental retinal vascular disease [27,43].

An important anatomical difference is the absence of a macular area in the porcine retina. However, the porcine retina has a cone-rich area, the visual streak, which is comparable to the human fovea [44]. The visual streak is horizontally oriented and located superior to the optic nerve head extending to the ora serrata [45,46] Klik eller tryk her for at skrive tekst. The visual streak consists of a relatively stable and high concentration of cone photoreceptors compared with other retinal regions [45,47]. Therefore, the porcine retina allows for studies of experimental RVO in a cone-rich area if the visual streak is affected by the induced occlusion.

Generating ischemia in experimental RVO is an important aspect. In order to induce ischemia observed as retinal capillary non-perfusion, our group recommends occlusion of two branch retinal veins. Occlusion of two branch retinal veins will result in retinal ischemia, as the porcine retina cannot compensate for occlusion of two adjacent retinal veins.

Earlier reports on experimental RVO are predominantly based on experiments with only a few animals. By including a series of studies conducted throughout a period of ten years, we show that laser-induced experimental RVO is a model with high reproducibility. Experimental BRVO is not associated with recanalization if sufficient laser is applied. Experimental CRVO is more difficult to induce, and it may take more animals to reach a sufficient sample size as occlusions must be balanced.

Overall, porcine models have a great number of advantages that make them superior as an animal model for experimental RVO. These include shared anatomical similarities with humans, great accessibility for the performance of surgical and diagnostic procedures, and easy availability [27].

### 4.2. Limitations of Experimental RVO in the Porcine Eye

RVO is particularly common in elderly individuals, and more than 50% of patients with RVO are older than 65 years of age at onset [48]. Arterial hypertension, systemic arteriosclerotic vascular disease, diabetes mellitus, and hyperlipidemia are important risk factors for RVO [49]. Experimental models of BRVO and CRVO are limited by the fact that young and healthy animals are used for the experiments. Young animals with healthy cardiovascular status are likely to recover faster than elderly humans with underlying arteriosclerosis following several years of hypertension. Based on angiographic findings in our studies, we have previously observed that pigs compensate fairly well when a single retinal vein is occluded [29,30,31].

While the porcine eye is well suited for studies of experimental RVO in terms of size and vascularization, the use of porcine eyes has a number of limitations. Compared to smaller animals, such as rodents, pigs carry a greater risk of infection, therefore demanding much more care and expenditure [50]. Monkeys would be the most suitable animals for studies of experimental RVO, as the monkey retina has a macular region with a foveal area. However, monkeys can be highly expensive, and permission to use monkeys as experimental animals may be difficult to obtain. Furthermore, many species face a high risk of becoming extinct [45].

### 4.3. Alternative Models of Experimental RVO

Several models of experimental RVO are available. Model selection should be based on the objective of the research and the available laboratory facilities. Laser-induced RVO can be performed using intravenous injection of photosensitive dye, which absorbs the laser and reduces the number of applications and the amount of energy needed. For protein studies, the use of photosensitive dye is generally not recommended, as radiation of the dye may cause protein modifications that make the assessment of the results difficult [27,34].

Photodynamic therapy (PDT) can be used to induce RVO, with the applications being made directly onto the retinal veins after intravenous injection of photosensitive dye. PDT-induced RVO has a moderate-to-high success rate for achieving occlusion [27]. PDT-induced RVO may be complicated by retinal necrosis if performed using unfavorable settings. As PDT-induced RVO may be associated with protein modifications due to the use of dye, the method is not well-suited for proteome studies [27,34].

Diathermic cauterization is an intravitreal procedure, which can induce both experimental BRVO and CRVO by applying diathermy on selected retinal veins [27,40]. Diathermic cauterization has a high success rate for achieving occlusion. Nonetheless, diathermic cauterization is an invasive procedure, which may affect the vitreoretinal interface and its proteome [27,34].

Transient clamping of the optic nerve is an invasive procedure, which can induce experimental CRVO and has a high success rate for achieving occlusion. However, transient clamping of the optic nerve also affects the ciliary vessels and central retinal artery, thereby not creating an isolated CRVO [27].

Intravitreal injections with substances such as thrombin can induce experimental BRVO and CRVO without any mechanical vascular damage. Nonetheless, intravitreal injections with substances may affect both retinal arteries and veins and generate features unrelated to RVO. RVO induced using intravitreal injections is only supported by a limited number of studies [27].

### 4.4. Clinical Future Perspectives of Porcine RVO Models

Studies of experimental RVO have generated significant insights into the pathological mechanisms activated in RVO. The retinal molecular mechanisms of RVO are a key element in developing novel therapeutic modalities and improving existing treatments. Testing intravitreal anti-VEGF agents and dexamethasone intravitreal implants in experimental RVO may potentially generate knowledge which can be used to develop personalized treatment approaches. Individualized treatment may result in improved visual acuity as well as fewer injections and follow-up visits.

## 5. Conclusions

Laser-induced RVO is a useful model for the study of molecular changes at the retinal level. A workflow consisting of advanced proteomic techniques, Western blotting, and immunohistochemistry can bring important insights into the complex pathological mechanisms which are activated in the retina following RVO. Laser-induced BRVO is a highly reproducible method and recanalization of occluded veins is generally not observed if sufficient laser is applied. Experimental CRVO is technically more challenging to induce as it is important to induce only partial occlusions by displacing thrombotic material to the lamina cribrosa. Complete occlusion of all major branch retinal veins of the retina results in severe ischemia. Therefore, successful experimental CRVO requires a balanced approach. The porcine eye has a number of advantages due to its similarities with the human eye, including similar size and a fully vascularized retina. Understanding molecular mechanisms through studies of experimental RVO and tests of intravitreal therapies may potentially generate knowledge which can be used to develop individualized treatment approaches for the benefit of patients with RVO.

## Figures and Tables

**Figure 1 medicina-59-00243-f001:**
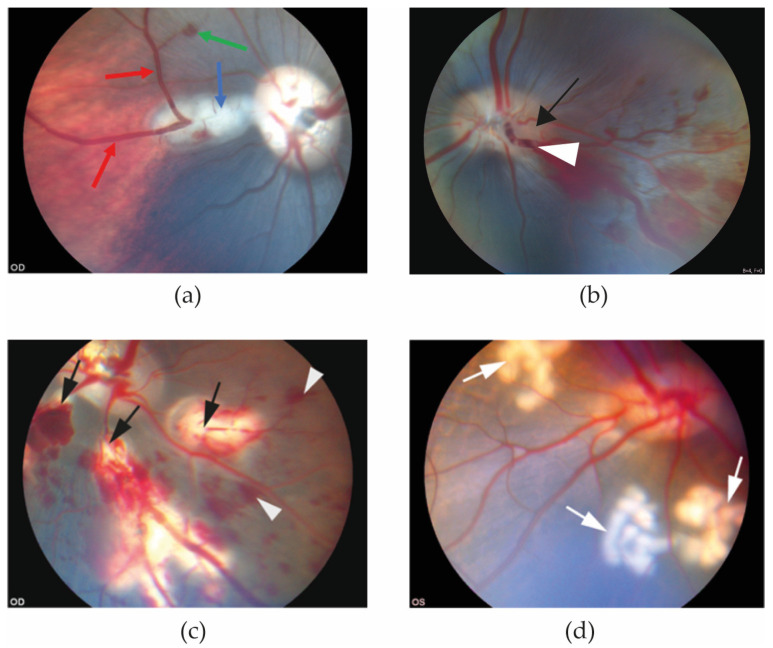
Funduscopic images obtained 20 min after RVO was induced. (**a**) Eye with experimental BRVO. Retinal hemorrhage (green arrow) and dilation of retinal veins (red arrows) are observed upstream of the site of the occlusion (blue arrow) [25]. (**b**) Experimental CRVO can be created by application of laser around a retinal vein at the border of the optic nerve head (black arrow) followed by application of laser directly on the vein, displacing thrombotic material towards the lamina cribrosa. (**c**) Experimental CRVO can also be created by complete occlusion of 3–4 branch retinal veins, creating a condition in which the entire retina is affected by occlusion. Complete occlusion of 3–4 branch retinal veins results in severe ischemia constituting a severe ischemic CRVO. Laser-induced occlusion (black arrows) resulted in dilation of the occluded veins upstream of the occlusion sites. Flame-shaped hemorrhages (white arrowheads) developed shortly after the vein occlusions were induced [32]. (**d**) Control eye. Control eyes can be created by generating areas of laser applications similar to the RVO eye, using the same amount of energy and number of applications as in the RVO eye (white arrows), but without inducing occlusion. This control eye serves as control eye for (**c**) [32].

**Figure 2 medicina-59-00243-f002:**
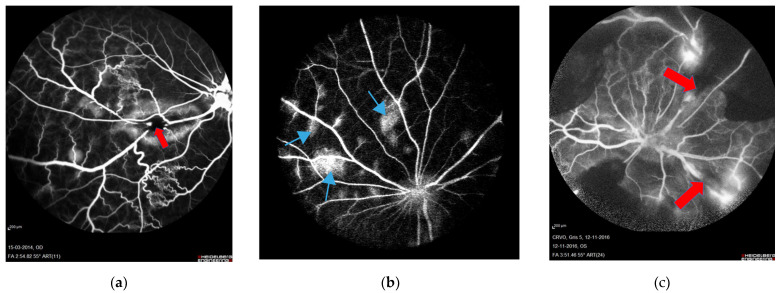
Fluorescein angiography performed four days after RVO. Sites of occlusion are marked with red arrows. (**a**) BRVO resulting in leakage and shunting to nearby veins. Retinal capillary non-perfusion is not observed [34]. (**b**) CRVO created by displacement of thrombotic material by laser towards the lamina cribrosa. Leakage is observed following CRVO (light blue arrows). This model corresponds to non-ischemic CRVO. (**c**) CRVO with complete occlusion of four branch retinal veins, resulting in retinal capillary non-perfusion and no sign of recanalization [32]. This model corresponds to ischemic CRVO.

**Figure 3 medicina-59-00243-f003:**
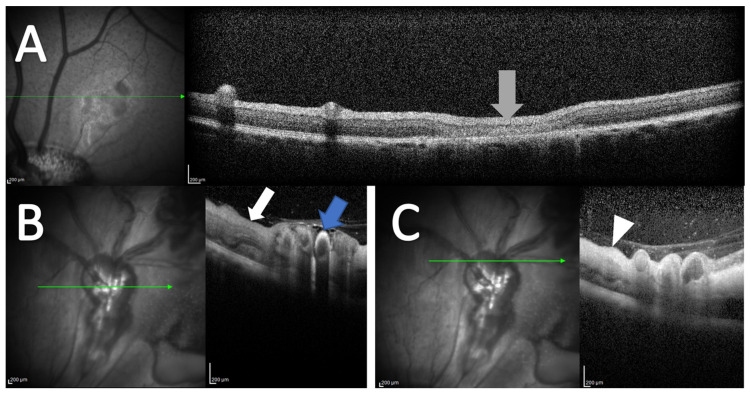
OCT following experimental CRVO. (**A**) OCT of control eye. No retinal thickening is observed. Retinal thinning is observed in areas where the control laser is applied (grey arrow). (**B**,**C**) OCT obtained four days after induced CRVO. (**B**) Hyperreflectivity is observed in the wall of the occluded retinal vein (blue arrow). Retinal thickening is observed in the area drained by the occluded vein (white arrow). (**C**) Retinal thickening is observed in the retinal area affected by the occlusion (white arrowhead). OCT scans are from an animal included in [32], but the OCT scans were not published previously.

**Figure 4 medicina-59-00243-f004:**
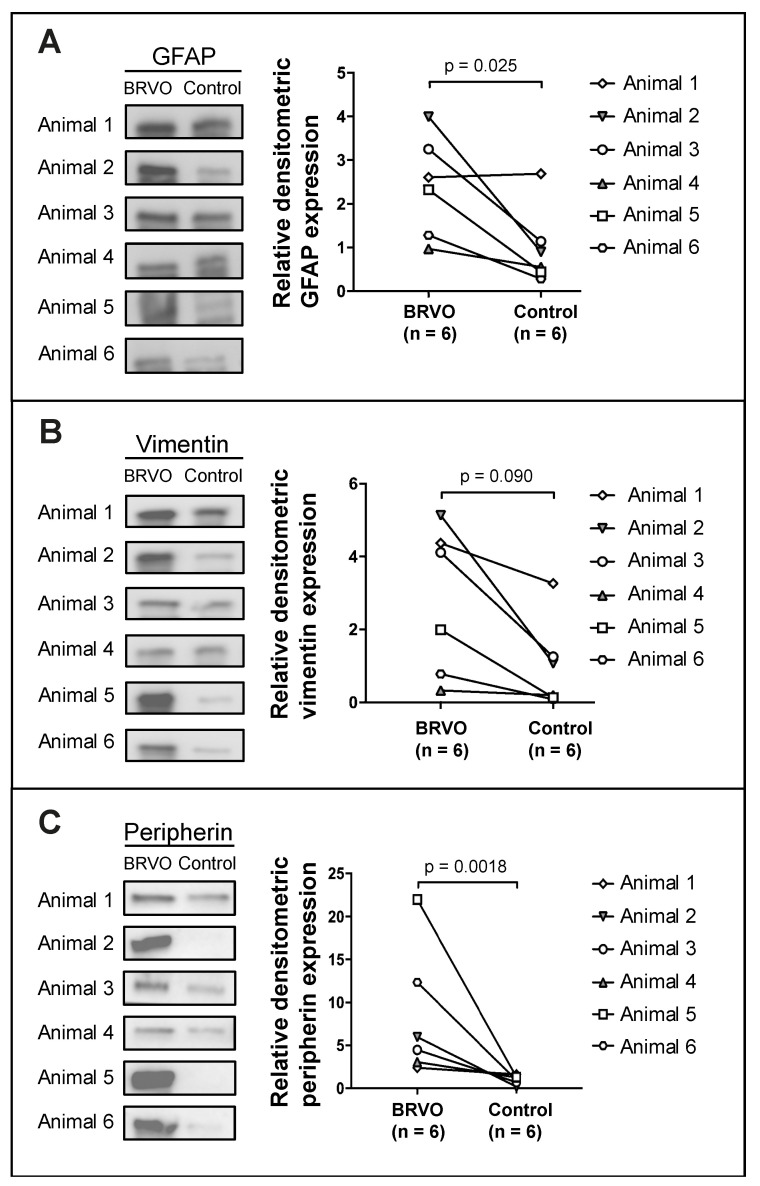
Western blots from a study of experimental BRVO in which discovery proteomics identified increased levels of Müller cell proteins in BRVO, including GFAP, vimentin, and peripherin. (**A**) GFAP, (**B**) vimentin, and (**C**) peripherin were verified using Western blotting, confirming increased levels of GFAP (*p* = 0.025) and peripherin (*p* = 0.0018). Elevated levels of vimentin were observed, but not found to be significant. Subsequently, targeted mass spectrometry using SRM was performed to validate the increased content of (**B**) vimentin, confirming the increase in vimentin in BRVO (fold change = 2.45; *p* = 0.00021). Based on our experience, SRM may be expected to be more sensitive than Western blotting, which may be considered a semi-quantitative method. A paired sample t-test was performed on log10-transformed densitometric data. Western blotting was performed on samples from [25], but the Western blots were not previously published.

**Figure 5 medicina-59-00243-f005:**
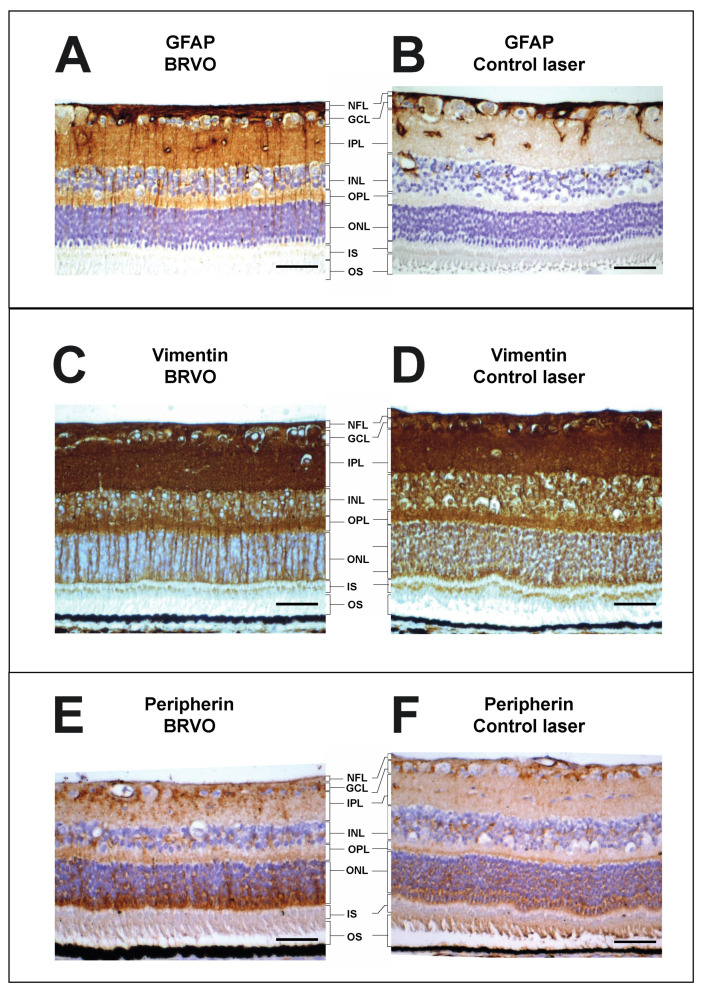
Immunohistochemistry of BRVO and control. (**A**,**B**), immunohistochemistry showed an upregulation of GFAP in the retinal ganglion cell layer, inner plexiform layer, and outer plexiform layer following aflibercept intervention in BRVO. (**C**,**D**), immunohistochemistry showed an upregulation of vimentin in the retinal inner plexiform layer and outer plexiform layer following BRVO compared to control. (**E**,**F**), immunohistochemistry showed an upregulation of peripherin in the retinal ganglion cell layer, inner plexiform layer, and outer nuclear layer following BRVO compared to control. Scale bar = 31 μm. Reaction color: brown. ILM: inner limiting membrane; NFL: nerve fiber layer; GCL: ganglion cell layer; IPL: inner plexiform layer; INL: inner nuclear layer; OPL: outer plexiform layer; ONL: outer nuclear layer; IS: inner segment; OS: outer segment. The samples were collected in [25], but the stainings have not previously been published.

**Table 1 medicina-59-00243-t001:** Table showing the number of eyes with experimental BRVO and CRVO used in the various studies.

	Eyes (n)	Animals (n)	References
BRVO	65	38 *	[25,29,30,31,33]
CRVO	43	27 **	[26,30,32]
Controls	22		[25,26,31,32]
Total	130	65	8

* A total of 11 control eyes was obtained from these animals. ** A total of 11 control eyes was obtained from these animals.

**Table 2 medicina-59-00243-t002:** Pros and cons of Göttingen minipigs and Danish Landrace pigs.

Animal	Pros	Cons
Danish Landrace pigs	Eyes are easily enucleated.	Adult animals weigh up to 200 kg.
	Relatively cheap (10% of cost of a Göttingen minipig).	Not well suited for long-term studies > 2 months.
		Expenses per animal are high compared with rodents.
Göttingen minipigs	Well suited for long-term studies. Adult animals weigh up to 40 kg.	Deep orbital cavities which can make enucleation challenging.
		Expensive.

## Data Availability

The data are published as supplementary files of references [25,26,29,30,31,32,33].

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
