# Peer review of "Laser-Induced Porcine Model of Experimental Retinal Vein Occlusion: An Optimized Reproducible Approach"

_medicina, 2023, doi:10.3390/medicina59020243_

Round 1

Reviewer 1 Report

In this paper, Maeng et al. reviewed laser-induced porcine model of experimental RVO based on own lab studies. Overall, this is a well-written paper with nice flow throughout the manuscript. I have few comments, which I hope can further improve an already good paper:

1. Introduction, lines 35-36. It is true that RVO is the second most frequent retinal vascular disease. It is also one of the most prevalent causes of macular edema and maculopathy. This fact deserves to be mentioned. 

2. Regarding the introduction, as this paper deals with pathophysiology in general, it would be a benefit to briefly highlight demographics of RVO and known/established risk factors. 

3. Regarding Section 2.1., the authors recommend the Danish Landrace over the Göttingen pig. It would be nice to see a table of pros and cons for Göttingen vs. Danish Landrace for better overview and to understand the basis of their recommendation.

4. Line 122: Regarding "On the contrary", this sentence does not contradict anything. Please rephrase.

5. Section 2.4. Regarding sample collection, please elaborate on the quality of the samples and the impact on the analyses, if the eyes are stored at -80 C and thawed prior to dissection.

6. Regarding 4.1. and 4.2. Strenghts/limitations, would it be possible to list various animal models of RVO and compare pros and cons for comparison and overview?

Reviewer 2 Report

The authors provided an interesting and detailed description of RVO in the porcine eyes. The contents are very informative. I have some minor concerns and would like the authors to address:

1, In the method section 2.2.1 and 2.2.2, laser spots were applied around the vein to narrow the vessel. Please indicate the space between the laser spots or if the spots were closely contiguous with each other. Please illustrate the intensity of the laser reaction (whitish, snow-whitish etc).

2, The authors mentioned that a successful experimental CRVO should be balanced and partial, while in the results section, both Fig 1b and Fig 2b exhitied CRVO induced by laser in the four retinal vein branches, resulting in prominent ischemia. Could the authors provide some CRVO imaging (fuduscopy and FA) showing laser near the optic disc margin inducing the "incomplete occlusion"?

2, 
